# Magnetic to the Core – communicating paleomagnetism with hands-on activities

Annique van der Boon[1, 2], Andrew J. Biggin[1], Greig A. Paterson[1], Janine L. Kavanagh[1]

[1]Department of Earth, Ocean and Ecological Sciences, University of Liverpool, Jane Herdman Building, 4 Herdman Street, Liverpool, L69 3GP, UK
[2]Now at Centre for Earth Evolution and Dynamics, University of Oslo, ZEB building, Sem Sælands vei 2A, 0371 Oslo, Norway

*Correspondence to:* Annique van der Boon (avanderboon.work@gmail.com)

**Abstract.** Paleomagnetism is a relatively unknown part of Earth sciences that is not well integrated into the school curriculum in the United Kingdom. Throughout recent years, there has been a decline in the number of Earth science students in the UK. In 2018 and 2019, we developed outreach activities and resources to introduce the scientifically-engaged general public to paleomagnetism and raise awareness of how geomagnetism affects society today, thus putting paleomagnetism, and Earth sciences, in the spotlight. We tested our ideas at local events that were visited mostly by families with small children, with tens to hundreds of participants. Our project culminated in the 'Magnetic to the Core' stand at the Royal Society Summer Science Exhibition in 2019, which is visited by members of the general public as well as students and teachers, scientists, policymakers and the media. At this event, we communicated the fundamentals of paleomagnetism through hands-on activities and presented our recent research advances in a fun and family friendly way. To test the impact of our exhibit on knowledge of paleomagnetism and Earth's magnetic field on visitors, we designed an interactive quiz and collected results from 382 participants over 8 days. The results show a significant increase in median quiz score of 22.2% between those who had not yet visited the stand and those who had visited for more than 10 minutes. The results from school-age respondents alone show a larger increase in median score of 33.5% between those who had not yet visited and those who had spent more than 10 minutes at the stand. These findings demonstrate that this outreach event was successful in impacting visitors' learning. We hope our Magnetic to the Core project can serve as an inspiration for other Earth science laboratories looking to engage a wide audience and measure the success and impact of their outreach activities.

## 1 Introduction

Paleomagnetism is the field of research that deals with reconstructing Earth's magnetic field in the past, as recorded by rocks. Paleomagnetism is an important field of study through its relevance to the ongoing geomagnetic protection of Earth from space weather. Severe space weather is named by the UK government as one of the major risks to society (National Risk Register, 2020). Paleomagnetism is also important for its utility in solving geologic problems, and has provided some of the first independent evidence for the theory of plate tectonics in the 1960's. Important contributions were made to the emerging field of paleomagnetism in the UK in the 1950's (Merrill et al., 1998). There is, however, little awareness of paleomagnetism and its uses and benefits amongst the UK public.

Paleomagnetism is a truly multi-faceted field as it bears upon all the natural sciences; it provides excellent possibilities for contributing to teaching in schools with links to physics (magnetism), chemistry (composition of magnetic materials), mathematics (trigonometry and vector calculus), geography (plate tectonics), and biology (magnetotactic bacteria). At its heart, paleomagnetism is a geoscience discipline, incorporating geophysics, geology and physical geography. While geography is taught widely in UK schools, the availability of geology is much more patchy and the number of courses is declining (Boatright et al., 2019). The number of A-level students roughly halved since the 1980's (peak at nearly 4000 entries), with a recent drop from 2500 to 1500 from 2014 to 2018. Geophysics is not taught until university level, although plate tectonics and other aspects of Earth science are taught within the national curriculum under geography and science. From 2017 to 2019, the number of geology students at UK universities decreased around 10% yearly (Boatright et al., 2019), thus the need to increase awareness of the importance of Earth sciences to society among school students and their parents is increasingly pressing. Geophysicists are named on the list of shortage occupations in the UK (UK government website). Although paleomagnetism outreach happens on local, regional and national levels, there is a scarcity of documentation on particular paleomagnetism outreach events or activities (Ayala et al., 2021), and it is thus difficult to assess what paleomagnetism outreach is being done around the world and how effective it is. Outreach is increasingly valued as an activity in which scientists should participate, and we show here that, with relatively little equipment, audiences can learn a lot about Earth's magnetic field.

## 1.1 The Royal Society Summer Science Exhibition

The Summer Science Exhibition of the Royal Society dates back to 1778, when the president of the Royal Society started 'conversaziones', in which fellows of the Royal Society could talk about their scientific work to members of the public. These events developed into the Summer Science Exhibition, which has been running in its current form for more than 30 years (Royal Society website). The exhibition is held in the building of the Royal Society in central London, close to Buckingham Palace. It attracts thousands of visitors each year, and runs for 7 days, at least 8 hours each day, and hosts 22 stands. Teams on a stand usually consist of 20-50 people, depending on the budget and number of collaborators. Often teams put in proposals for a stand as a collaborative undertaking, with multiple universities or institutes involved in a bid. In total, around 700 scientists present their work at the exhibit each year. For the 2019 edition, about a third of the proposals were successful, and our stand was the only dedicated Earth science at the exhibition, as well as the only one from the University of Liverpool.

## 2 Magnetic to the Core

We describe here the development of our paleomagnetism outreach activities and resources, the reactions from participants, and our evaluation process and its results. Ultimately, these outreach activities formed part of the University of Liverpool's Research Excellence Framework (REF) 2021 submission, highlighting its perceived value as a professional activity. Our activities can easily be adapted and emulated to fit other countries, labs and audiences. We present this project in the hope that it can serve as an inspiration for other groups to perform outreach with impact in paleomagnetism and Earth sciences.

## 2.1 Outreach Team and Training

We formed a team that was diverse in terms of age, career stage and gender. The team consisted of PhD students (7, of which 2 women), postdocs (7, 4 women), with the addition of MSc students (2, both women), as well as a technician (man), a research fellow (man), senior lecturers (2, both women) and professor (man). Actions such as using name badges with the option to include preferred pronouns would be one way at future events to publicly recognise gender as non-binary and therefore increase the inclusivity of the event with respect to gender identity. Unfortunately, the lack of racial diversity in Earth sciences (Bernard and Cooperdock, 2018; Dowey et al., 2021; Dutt, 2019), and paleomagnetism in particular (Ayala et al., 2021) was also reflected in our team. In order to try and recruit team members from non-white backgrounds, we contacted other paleomagnetic laboratories in the UK several months before the exhibition, and tried to enthuse (PhD) students and researchers to join our team. Unfortunately this yielded no results in terms of increased racial diversity, and we were thus unable to form a team that showed the same diversity as the UK society and the audience at the exhibition. This is problematic as when young people do not see paleomagnetists (or scientists in general), who look and sound like them, this gives the indirect impression that paleomagnetism (or science) is not for them. Evidenced action lists to address the lack of diversity in geosciences, including marginalization related to gender identity, ethnicity, sexuality, physical ability, socio-economic background and intersectionality, should be consulted when putting together teams that deliver outreach activities and the way the team and the resources they create are presented (e.g. Ali et al., 2021; Dowey et al., 2021; Vander Kaaden et al., 2021; Núñez et al., 2020).

We developed our outreach activities over several months, and tested our activities at smaller outreach events that attracted tens to hundreds of visitors, mostly families with young children. These smaller events were a good way to obtain some first experience in outreach for most team members, and allowed us to judge which activities were most engaging. It was also a good way to form ideas about the knowledge levels of visitors and adapt our outreach to that, as well as try different strategies to measure the impact of our outreach.

Because of differences in backgrounds and level of experience in paleomagnetism and outreach amongst our team members, we organised a day-long training session for everyone on the team for the Royal Society Summer Science Exhibition. This training session was, for a large part, based on the training day that the Royal Society provided for 2 members of each stand, which was given by the Science Museum Group. The training was adapted to provide a more stand-specific training for our Magnetic to the Core team.

During the training, we invited the university press officer and social media officer to give presentations on how to talk to journalists and use social media during the exhibit. We developed a list of commonly used jargon in paleomagnetism and came up with other terms that would be better suited to explain our science to people without a scientific background (see Table 1). We thought about what different kinds of visitors would like to take away from the experience and designed interactions suited to specific groups of visitors (kids, families, groups of friends, independent adults, etc.). We then role-played short (2-3 minute) interactions with different groups of visitors so our team members became proficient in engaging in a fun way with visitors, while also bringing across the key messages of our stand. The interactions were designed using a framework for engagement that consisted of four phases. The first phase is the 'hook', in which we drew the attention of a

visitor and tried to get them engaged (e.g. asking 'would you like to do an experiment?'). Subsequently, we would 'inform' the visitor, by introducing ourselves and showing or explaining them something. Then, we would 'enable' the visitor, often by having them do a hands-on activity. Finally, the interaction would end with

'extending' the experience, in which we would give them something to take home, or show how what they had learned linked to their daily lives. This framework was particularly useful because it showed a clear beginning and end to interactions, and enabled us to bring across our key messages in an engaging way, while only taking a few minutes. The framework made it easy for visitors to walk on to the next stand after a few minutes, or stay longer if they were interested in knowing more.


| Scientific jargon | Alternative terminology |
|---|---|
| (geo)dynamo | magnetic field (generator) |
| sample | piece of rock |
| specimen | piece of rock |
| paleomagnetism | study of the magnetic field in the past |
| magnetometer | machine that measures a magnetic field |
| modelling | computer modelling |
| simulation | computer model |
| magnetostratigraphy | record of the magnetic field in the past |
| magnetic susceptibility | how magnetic something gets |
| remanence | magnetic memory |
| reversal | magnetic pole flip/compass points the other way |
| polarity | direction (north/south) the compass needle points |
| (outer) core | engine of the magnetic field inside the earth |
| mantle | inside of the earth |
| dipole | bar magnet |
| quadrupole | complex magnet |
| space weather | particles from the sun |
| mineral | particle in a rock |
| anomaly | weird spot |

Table 1: Scientific jargon and plain language alternative terminology

During the training, we discussed good and bad interactions, and practiced having a two-way conversation with visitors. We practiced encouraging visitors to ask questions, instead of providing them with information in a

one-way interaction. We also practiced and discussed difficult scenarios (see supplementary file 1), to ensure the safety of both our visitors and team members. The scenarios we practiced involved interactions that did not go as planned, so each team member would know what to do when things (or interactions) went wrong, and who to ask for help.

**2.2 Scientific content and learning outcomes**

We developed a range of activities that were centred on the research tools of paleomagnetists and invited visitors to experience what it is like to be a paleomagnetist. The tasks were related to some fundamental

paleomagnetic work, and the team members could easily form links to their own research when interacting with visitors.


We trained our team in communicating science to lay audiences, where we focused on three key messages:

      1. Earth has a magnetic field that acts as an invisible shield that protects us from space weather.

      2. Paleomagnetism is the study of the ancient magnetic field in rocks and anyone can do it.

      3. Paleomagnetism tells us about how the inside and outside of Earth have changed in the past, and

how they might change in the future.

Interactions were designed to get one or more of these key messages across via learning outcomes. The learning outcomes are the messages that we wanted our audience to realise and remember after visiting our stand. The four learning outcomes were:

1. Rocks are cool and they can record magnetic fields.

      2. The magnetic poles of Earth reverse.

      3. Different materials have different magnetic properties.

      4. Paleomagnetism is practical, hands-on, and anyone can do it.

Defining learning outcomes enabled evaluation of the outreach activity, and we designed our hands-on activities

so that they all related to one or more learning outcomes (see Table 2). The use of learning outcomes allowed us to test whether visitors had learned something new about paleomagnetism from visiting the stand.

**2.3 Hands-on activities**

There was a logical order to the activities in our stand (see Figure 1), although they could all be enjoyed

separately without previous knowledge acquired through the other activities or prior knowledge of paleomagnetism. The logical order that we chose follows the path that a student who applies paleomagnetism in a research project usually takes. 1) Without a lot of prior knowledge on paleomagnetism, students go on fieldwork to collect samples. 2) Students then measure samples to determine the direction of Earth's magnetic field when the rock formed and learn to interpret the results. 3) Students do additional rock magnetic

measurements that give information on the magnetic minerals present in the samples. 4) Students then go on to learn more about the fundamentals of paleomagnetism to put their results into a wider scientific context. Most of the hands-on activities benefit visual and kinesthetic learning preferences (https://vark-learn.com/). The interaction with team members at the stand catered for visitors with an aural learning preference (who prefer discussions and stories). The backdrop of our stand catered to visitors who had a learning preference for reading

(see also Table 2).

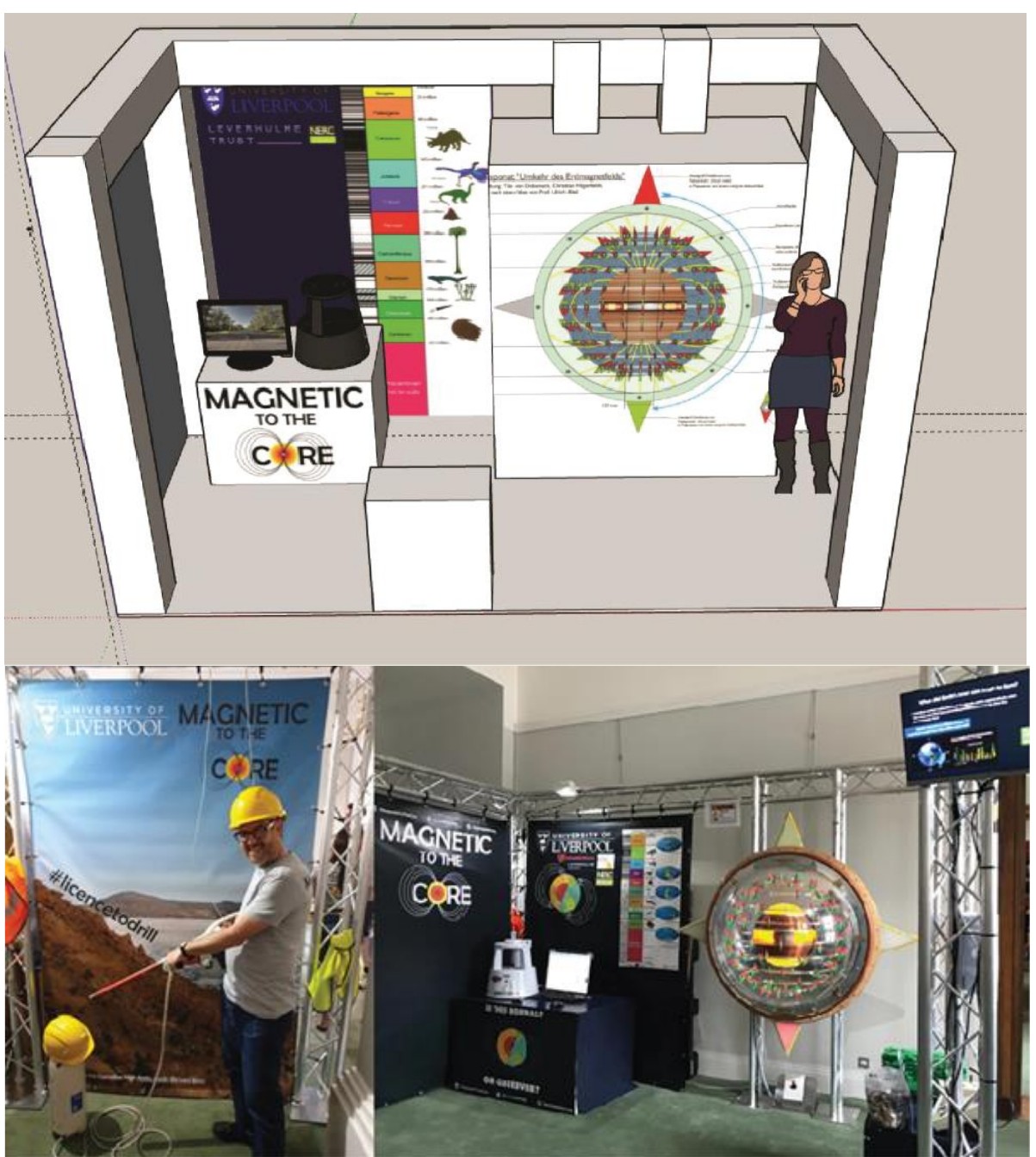

**Figure 1. Top: Preliminary design of the stand in Sketchup, size of the entire stand is 4x2 metres. Bottom left: Andy Biggin getting his 'license to drill'. Bottom right: The Magnetic to the Core stand at the Royal Society Summer Science Exhibition, 2019. On the left, the Kickstool Magnetometer; in the centre, the magnetic globe; on the right, Rock or Choc. License to drill is on the far left of this image, behind the Magnetic to the Core banner.**

### 2.3.1 Experiencing sample collection:

Paleomagnetic studies often start out with sample collection. In order to translate taking paleomagnetic samples in the field to an activity that could be done indoors, we had a gasoline-powered drill with a printed background of rocks in the high Arctic that was part of the research conducted by a team member (Bono et al., 2013). For safety, the drill was drained of fuel and cleaned so it could not be turned on, and it was secured to the stand with

high-strength steel cables (see Figure 1). Visitors could then pose for a picture with the drill, dressing up in high-visibility vests and a hard hat, emphasising that safety is an important component to fieldwork.


This activity was tied to Learning Outcome 4, and demonstrated that paleomagnetism is practical, hands-on, and anyone can do it in the sense that taking samples in the field is a practical exercise that requires little previous knowledge. Fieldwork is a favourite part of the job for many paleomagnetists. This activity shows that science can be fun and can take you to some of the most stunning places on Earth.


### 2.3.3 Measuring samples:

The visitors could then go on to measure samples that they had just 'taken'. We provided a set of lavas from Iceland, which visitors could measure on the kickstool magnetometer (see Figure 2). This magnetometer, designed in collaboration with Magnetic Measurements (http://www.magnetic-measurements.com/), was built

specifically for outreach, with the name coming from the low cost kickstool that forms the main frame of the system. The magnetometer is connected to a screen that shows the polarity of the lava samples. Half of the samples had a normal polarity (i.e., corresponding to an Earth magnetic field in which the magnetic poles are in the same position as today), while the other half of samples had a reverse polarity (i.e., the magnetic poles were in a reversed position from today's field).


This activity was tied to Learning Outcomes 1, 2 and 4. Visitors could hold the rocks, place them in the sample holder, operate the magnetometer themselves and deduce the polarity of the rocks. We explained to visitors during this activity that the last time Earth's magnetic field was reversed was around 780 ka, so the rocks for which they measured a reverse polarity must be at least 780,000 years old.


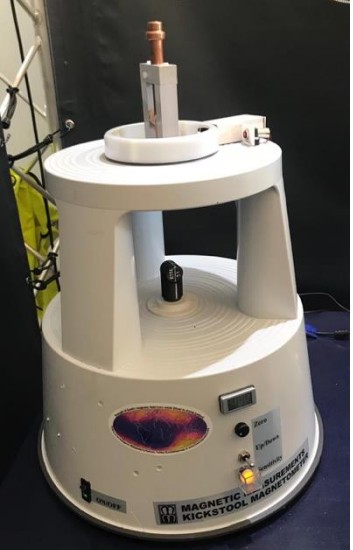

**Figure 2 - The Kickstool Magnetometer**

### 2.3.3 Rock or Choc:

'Rock or Choc' proved to be the most popular activity, as it was open to all ages, there was a gaming element to it, and visitors were given a chocolate pebble as a reward for participation. Using the concept of magnetic

susceptibility, which demonstrated Learning Outcome 3 (different materials have different magnetic properties); visitors could distinguish real rock pebbles from chocolate pebbles (see Figure 3). The real rock pebbles were black decorative basalt pebbles that are often used in aquariums, and have a high magnetic susceptibility. We gave visitors a set of pebbles in clear plastic boxes that could not be opened, and asked them to distinguish the real pebbles from the chocolate pebbles. Half of the pebbles were real rocks, the other half were chocolate. Visitors could try to distinguish them by visual inspection, weighing the boxes in their hands, and shaking the boxes. Although the weight of the real pebbles was slightly larger than that of the chocolate pebbles, it was nearly impossible to distinguish them. Visitors could then check if they had guessed correctly which pebbles were real rocks by placing them in a Bartington MS2 susceptibility meter. Because rocks contain magnetic minerals, they have a high magnetic susceptibility, and gave high values on the susceptibility meter. Chocolate is not magnetic, so the chocolate pebbles gave magnetic susceptibility values of around zero. As a reward for participating, we gave visitors a chocolate pebble that they could eat at the end of this activity. As part of our signage, appropriate allergen information was provided. Alternatively, this activity can be carried out using a compass, which will show deflection of the needle when brought close to a magnetic rock pebble. The chocolate pebbles will not show a deflection of the compass needle.

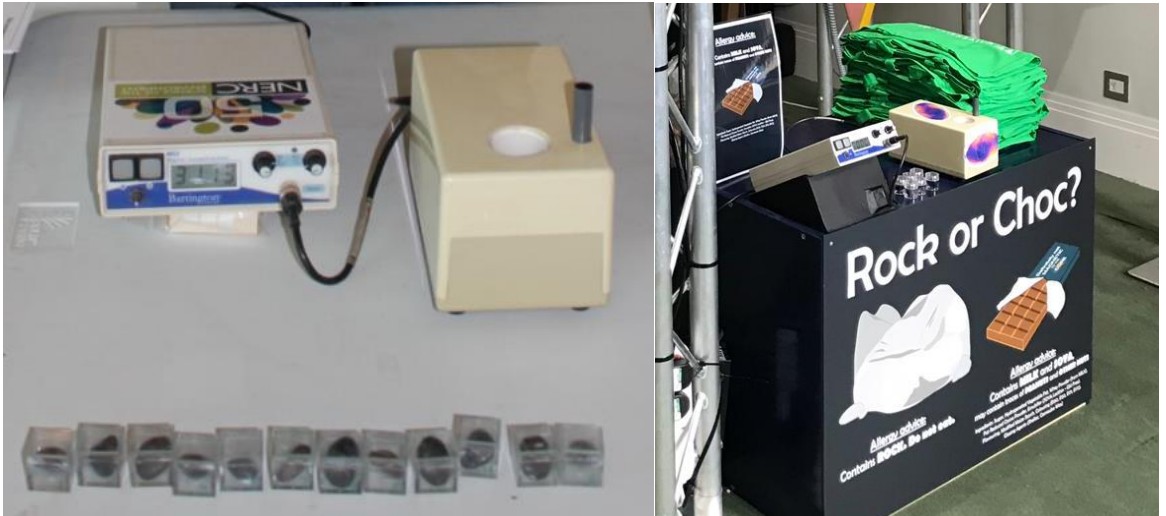

**Figure 3: Rock or Choc setup, with boxes of chocolate and rock pebbles, and a Bartington susceptibility meter.**

### 2.3.4 Magnetic globe:

Next, visitors could learn about the reversing magnetic field by simulating a polarity reversal in a scaled model of Earth's magnetic field and dynamo (see Figure 4). This model was borrowed from the paleomagnetic laboratory at the University of Bremen, who designed and built the globe. The globe contains a solid iron ball in the middle, which represents the solid inner core. Around this iron ball is a copper coil, which generates a magnetic field when a current is running through it. This coil represents the liquid outer core of the field, which is where Earth's magnetic field is generated. The mantle was represented by empty space, and small magnetic compasses were placed at a scaled distance to represent the Earth's crust. The globe could simulate a reversal by flipping a switch on the controller, which made the field go from a dipolar state to a quadrupolar state. This was visualised by the magnetic compasses, which changed orientation in response to the changing field. Pushing the

switch further made the field flip from a quadrupolar state back to the dipolar state in the opposite polarity as before. This activity was tied to Learning Outcome 2 (the poles of Earth's magnetic field reverse).

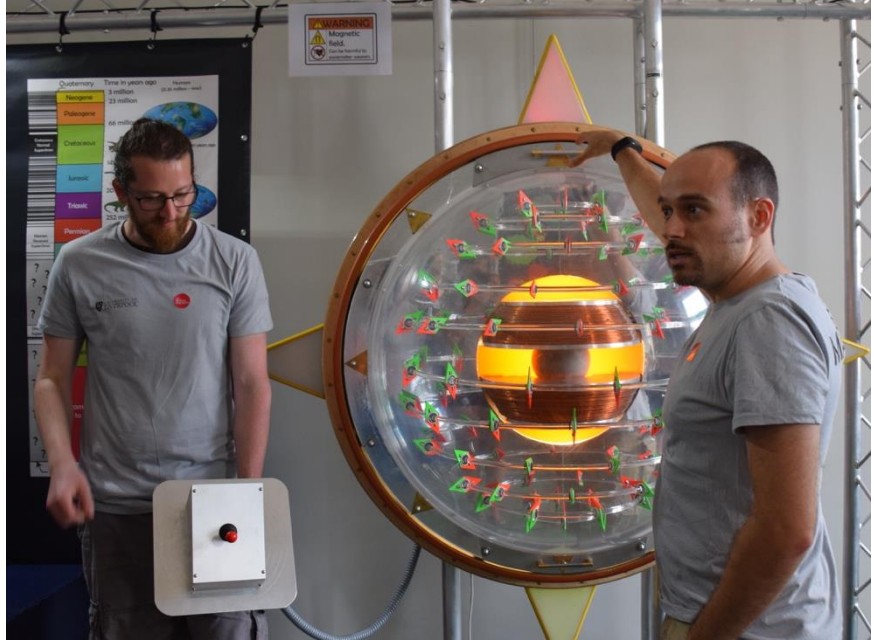

**Figure 4: Daniele Thallner and Greig Paterson with the magnetic globe and switch**

250

| Name of activity | Key message | Learning outcome | VARK |
|---|---|---|---|
| Licence to drill | 2 | 4 | V,K |
| Kickstool magnetometer | 2, 3 | 1, 2, 4 | K |
| Rock or choc | 2 | 3 | K |
| Magnetic globe | 1, 3 | 2 | V |

**Table 2: overview of relation between hands-on activities, key messages, learning outcomes and learning preferences (VARK; Visual, Aural, Read/Write, Kinesthetic)**

**2.4 Screen, posters, social media and freebies**

255 If it was busy at the stand, visitors sometimes had to wait for their turn to participate in the hands-on activities. To keep them engaged, we had a screen at the stand that showed summaries of each team member's research, as well as short videos of fieldwork. To extend the stand visit beyond the exhibition, we had two freebies for visitors (see Figure 5). One was a sticker with a map of the Earth and the magnetic field strength, with social media details of our stand and group. Another was a fridge magnet that shows a plate tectonic reconstruction of

260 Earth at 250 Ma (Torsvik et al., 2012) which could only be obtained through participation in the quiz. We also made printed booklets with ten facts about paleomagnetism, called 'Ten things you might not know about Earth's magnetic field' (van der Boon, 2019).

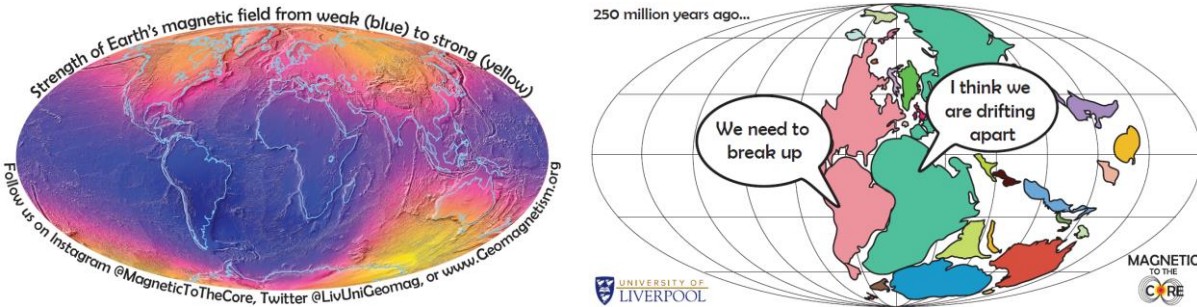

**Figure 5: Sticker (left) with links to websites (geomagnetism.org) and social media, and fridge magnet (right) that were handed out to visitors of the stand**

### 2.5 Quiz

To evaluate what visitors had learned from visiting our stand, we designed a multiple choice quiz, and we rewarded participation with a fridge magnet. The quiz consisted of nine questions about paleomagnetism and five questions related to visitors' background and their experience at the stand (see Supplementary File 2).

Trained stand exhibitors used a tablet and the QuickTapSurvey (https://www.quicktapsurvey.com/) app to evaluate the knowledge of people who had visited the stand for more than 10 minutes, less than 10 minutes, and those who had not yet visited the stand. The quiz was designed to assess the impact of the stand on visitors' ability to answer questions related to the learning outcomes. The quiz was a fun way to assess people's knowledge, and participants were generally very willing to take part. We also asked visitors whether they were in school, as this allowed us to roughly assess their age without collecting sensitive information. During the exhibition, there was one person on the stand at all times whose goal was to approach people to do the quiz. This team member would ask the questions to visitors and fill in the answers on a tablet. We trialled other approaches with questionnaires that visitors could fill in themselves at smaller outreach events, but this method was unsuccessful, as visitors were generally not very eager to fill in a questionnaire, or would only fill in half the answers.

### 2.5.1 Ethics

Ethical considerations were taken into account before gathering the data and the study was approved by an ethics panel at the University of Liverpool. Data collection was designed so that individuals could not be identified, and no sensitive information of participants was collected (following General Data Protection Regulations; GDPR). This however, limited the options for assessing the long term impact of the stand on participants, as a follow-up study of the same group of individuals was not possible.

### 2.6 Budget

Doing outreach events effectively can be an expensive undertaking. To make the costs of doing a large outreach event insightful, we provide the generalised budget for the Magnetic to the Core project (see Table 3), to help guide the plans of other groups with future outreach activities. We were able to secure a total of £24,000 for our stand. Most of the budget was provided by the University of Liverpool, and another significant part came out of research budgets of grants awarded to the research group from Leverhulme and the Natural Environment Research Council (NERC). Smaller amounts of funding were provided by the Royal Society and British

Geophysical Association. Four other grants to fund team members' participation in the exhibition, worth a total of £4,500, were unfortunately all unsuccessful. Our experiences here highlight the continued problematic nature of obtaining funding for outreach and public engagement, despite a professed eagerness from funding agencies to support these efforts and recognition of the significant gains that can be achieved. Scientists are often expected to develop and do outreach on top of their research and teaching, while outreach projects like Magnetic to the Core take a serious amount of time and dedication. Having a 2 months full-time postdoc leading the project was a bare minimum for the scale of this undertaking.

| Amount in | From | Amount out | What |
|---|---|---|---|
| £10,000 | University of Liverpool, School of Environmental Sciences | £5,500 | Costs for building stand and all materials |
| £10,000 | University of Liverpool, Faculty of Science and Engineering | £7,500 | Costs for accomodation and food for the team |
| £3,350 | Contribution from existing research grants (funded by Leverhulme Trust and NERC) | £5,000 | Costs for transport to and from London |
| £400 | Royal Society | £6,000 | 2 month full-time Postdoc salary |
| £250 | British Geophysical Association | | |
| £24,000 | | £24,000 | Total |

**Table 3: Indicative budget and costs of the Magnetic to the Core project**

Most of the costs of the project went into getting the 21 people on our team to London and accommodation there for 10 days. Team members stayed in shared accommodation booked through Airbnb and had an allowance of £15 per day for food. Another significant cost was the salary of the first author of this manuscript, who was employed as a postdoctoral researcher to manage the project full-time for 2 months. Although it is difficult to estimate the true cost that also takes into account any hidden costs, just the number of person-hours at the stand during the exhibition is around 450. This does not include any preparations, travel, build-up and taking down of the stand. We estimate that taking into account all hours that all team members worked in total would probably put the true cost at double the cost that our budget shows.

## 3 Results

Over the course of 7 days, the Royal Society Summer Science Exhibition, which featured 22 stands, attracted a total of 12,653 visitors. This included 1,518 students and 187 teachers from 89 schools.

### 3.1 Number of visitor interactions

We had several ways to get estimates of the number of visitors to the stand. One was through the number of freebies. We printed 1000 stickers, which were finished after day 4 of the exhibition. However, some groups of people would go on the hunt for freebies without visiting the stand, making this an overestimate of the number of visitors. To measure the number of visitor interactions at our stand specifically, the number of kickstool magnetometer measurements was used as a proxy. The results show that over the course of the week, 1,011 measurements were recorded (see Figure 6). Generally, participants measured between 2 and 6 samples, putting the estimate of the number of people using the kick stool magnetometer at between 170 and 500. Figure 1 shows the number of measurements performed per day. Not all visitors used the magnetometer, so the number of attendees to the stand as a whole was much higher. We registered 382 quiz responses, of which 165 had already visited the stand. In total, we estimate that at least several hundred and possibly one thousand visitors had interacted with our stand.

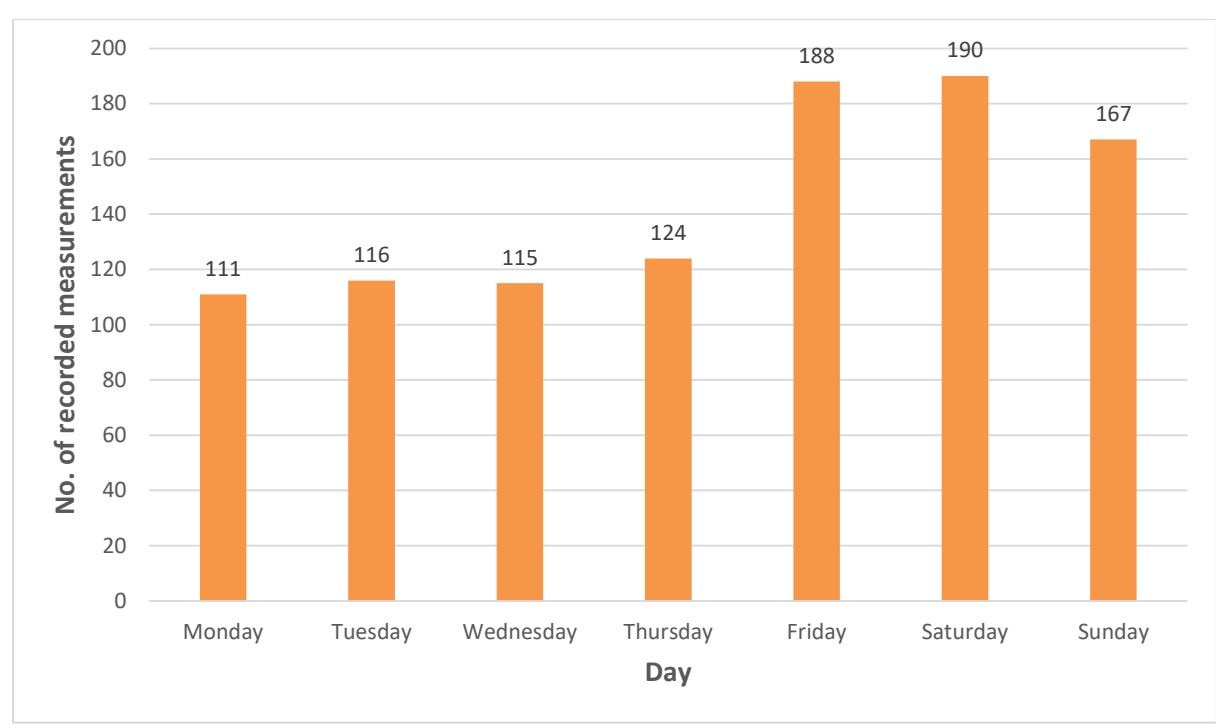

**Figure 6: Number of measurements performed on the kickstool magnetometer per day**

### 3.2 Impact on knowledge of visitors

The impact of this outreach event on the knowledge of visitors was measured by looking at the difference in quiz scores between participants who had and had not (yet) visited the stand. Because the data do not show a normal distribution, we use median values instead of averages. In total, there were 382 quiz responses recorded, of which 136 were by school-aged participants (see Table 4). As shown in Figure 7, the median quiz scores increased by 11.1% for those who visited the stand for less than 10 minutes compared with those who had not

yet visited the stand. For those who visited the stand for more than 10 minutes, their median scores increased by 22.2%.

| | Not visited | <10 mins | >10 mins | Total |
|---|---|---|---|---|
| All participants | 216 | 77 | 89 | 382 |
| School-aged participants | 66 | 36 | 34 | 136 |

**Table 4: Number of quiz responses**

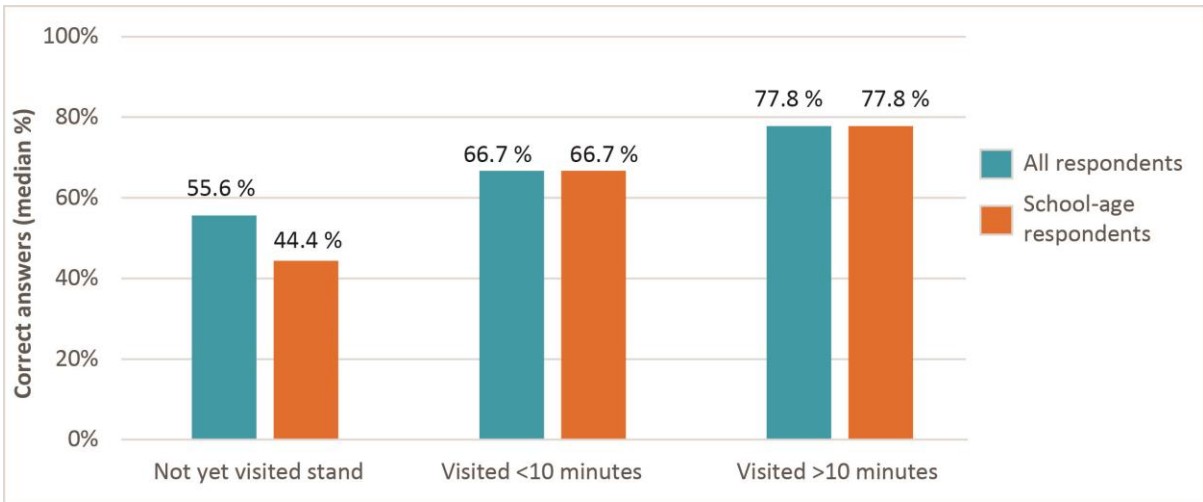

**Figure 7: Median quiz scores**


Comparatively, visitors of school-age who visited the stand for an extended period of time experienced a more significant increase in the accuracy of their scores in the quiz (see Figure 7). School-age participants who visited the stand for less than 10 minutes achieved a 22.3% increase in median scores compared to school-age participants who had not yet visited the stand. School-age participants who visited the stand for more than 10

minutes experienced a 33.5% increase in their median scores compared to those who had not yet visited. We use the Wilcoxon rank sum test for common medians (also known as Mann-Whitney test). Comparing the responses from people who had not yet visited the stand to people who had visited <10 minutes, yields Wilcoxon rank sum p-values of p=0.010 for all respondents, and p=0.003 for school-aged respondents. Comparing the <10 min responses to the >10 mins responses gives the following p-values for the tests: all respondents, p<<0.001,

school-aged respondents, p=0.004. In both cases, at better than the 0.5% confidence level, we can reject the null hypothesis that the responses from the different groups come from distributions with the same median. Thus, the differences are significant. Performing a 2-sample t-test (for equal means) on average scores (which ignores the non-normality of the data) also shows there is a significant increase in quiz score.

The answers to the question 'Can you tell us one thing you have learned from the stand' show that we were successful with regards to bringing across our learning outcomes. There are 210 answers to this question (see supplementary file 2), with 89 of these answers specifically relating to a learning outcome. Of these, 19 mention learning outcome 1 in some form (e.g. "Measuring Polarity With Magnetometer"), 61 mention learning outcome 2 (e.g. "Magnetic Reversals"), 7 mention learning outcome 3 (e.g. "Chocolate is not magnetic"), and 2 mention

learning outcome 4 (e.g. "That samples are taken by drilling"; see Figure 8). Another theme that is often mentioned is that the Earth's magnetic field is generated in the (liquid outer) core. Although, not specifically linked to a learning outcome, this was often discussed with visitors interacting with the magnetic globe. A common theme emerging from the answers not linked to a learning outcome is the deep Earth (e.g. "The Earth has a solid core", "Liquid outer core").


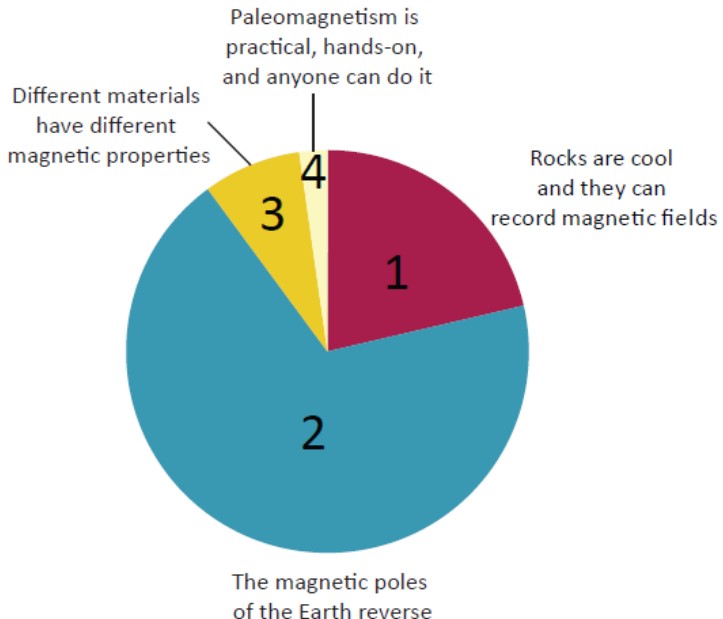

**Figure 8: Learning outcomes mentioned by participants as things they had learned**

**4 Discussion**

**4.1 Impact on knowledge of visitors**

Our results indicate that our stand had an impact on visitors' knowledge regarding paleomagnetism and Earth's magnetic field. One of the downsides of our approach, in which there was one team member who asked the questions and filled them in on a tablet, was that we were often dealing with groups of people, instead of individuals. Some groups wanted to do the quiz together as a team, and they would discuss amongst themselves which the most likely answer was. Other groups of people wanted to do the quiz as opponents, while it was only possible to fill in one answer at a time. The quiz thus only gives a rough indication of the impact of our stand on the knowledge of visitors. Furthermore, our question 'can you name one thing you learned from the stand', which was a good way to test if our learning outcomes had come across, often took a long time to fill in on the tablet. This led in some cases to incomplete answers, particularly when it was busy at the stand and people were queuing for the quiz. We trialled a different approach at the smaller outreach events, which was to let visitors fill this in themselves on a piece of paper. This, however, turned out not to be a good alternative, as visitors often left that box blank. Thematic analysis (Braun & Clarke, 2006) can be used to evaluate answers from this type of open question. Although Rock or Choc was extremely popular, visitors do not often refer to learning outcome 3, possibly because it is considered common knowledge. We do note that there are many answers along the lines of "Chocolate Can Look Like A Rock", but we did not count these in our analysis, because they do not specifically mention magnetism.

It is unclear if the targeted group is representative for society as a whole, as the prestige of this event and the location could have attracted attendees from more affluent backgrounds. The participants may represent a well-educated proportion of the general public, perhaps with prior general knowledge about science, thus making participants more familiar with scientific topics than the general population. The Royal Society arranged

participation for school groups, specifically aimed at schools from the least affluent areas to mitigate this for part of our studied demographic, the school-age participants.


Due to the barriers on collecting sensitive information (such as gender, age, contact details, etc.), we are unable to provide more specific details on the participants to the study. Our team members that were in charge of the questionnaires aimed to question an even spread of school-age participants and non-school-age participants, as well as people who had and had not yet visited the stand.


The adults who attended the exhibit were self-selecting, making it likely that they had a general interest in science prior to the exhibition. This is inferred by all respondents receiving a median score of 55.6%, despite not having visited the stand yet, and school-age respondents receiving 44.4% median scores when not having visited the stand yet. Therefore, collecting another dataset in a different location may be beneficial to see whether these

results are representative for the general public.

### 4.2 Benefits to the team

Because of the scale of the Royal Society Summer Science Exhibition, and the limited budget, we were required to create most of our resources ourselves. This meant that the entire team was heavily involved in creating the

exhibit. Many stands at the exhibition had larger budgets, which meant that parts of the work could be done by specialist companies for creating videos, designing and building stands and outreach materials, creating apps, etc. This was not an option for us because our budget limitations precluded us from outsourcing. Because of this, our team members gained a lot of experience in designing outreach activities, building exhibits, making videos, writing for lay audiences, etc. Talking to non-scientists about paleomagnetism was also a good training

in communicating fundamental scientific concepts in an accessible way. Team members reported feeling more confident in talking to lay audiences about their science after the exhibition. Science can be a secluded undertaking and team members felt very motivated by the enthusiasm for their science by the visitors of the stand.

The research group benefited from the exposure of our research, not only to the public, but also to policy makers, other scientists (including many Fellows of the Royal Society) and journalists. The University of Liverpool, who provided the bulk of the funding for the exhibit, along with the funding agencies benefited from widespread display of their logos at the stand and on associated websites and promotional materials (e.g., Figure 5).


### 5 Conclusions

We have presented our Magnetic to the Core project here in the hope that it will inspire other researchers to undertake Earth science outreach. Despite having a relatively small budget (£24,000) and little prior experience of major outreach activities amongst the team members, the Magnetic to the Core stand at the Royal Society

Summer Science Exhibition 2019 was successful in achieving its aims and provided tangible benefits to the team and funders, as well as the public who attended it. We have provided an example of how to measure the impact of outreach events, and share our budget and evaluation to demonstrate the actual cost and value of

outreach activities. By using the recordings from the kickstool magnetometer and quiz responses, we estimate that at least several hundreds of visitors interacted with the stand. We show that visiting the stand had a significant impact on visitors' knowledge of paleomagnetism and Earth's magnetic field through the increase in quiz scores. Increased amount of time at the stand increased the median scores further. In order to gain results that are representative of society as a whole, more data is needed to mitigate self-selection of participants. Overall, our experiences with Magnetic to the Core were positive, with every team member expressing their enjoyment participating. Although challenging at times, the events were rewarding and have strongly motivated us to continue our outreach efforts.

## 6 Acknowledgements

The Magnetic to the Core project was partly funded by Leverhulme Research Leadership award RL-2016-080, Natural Environment Research Council (NERC) standard grant NE/P00170X/1 (both to AJB), NERC Independent Research Fellowship to GAP (NE/P017266/1), with additional funding from the University of Liverpool, the British Geophysical Association and the Royal Society. This work could not have been done without all the hard work and enthusiasm of our team consisting of Anouk Beniest (VU Amsterdam), Richard Bono, Yael Engbers, Michael Grappone, Ben Handford, Louise Hawkins, Mimi Hill, Elliot Hurst, Tereza Kamenikova (University of Lancaster), Simon Lloyd, Domenico Meduri, Joe Perkins (Imperial College London), Georgia Quinn, Jenny Schauroth, Courtney Sprain and Daniele Thallner. We thank the team at the Royal Society and Amy Fry in particular for their help, enthusiasm and organisation. We benefited a lot from the training provided by the Science Museum. We thank all visitors of our stand for their interest in our research and their questions, the Paleomagnetic Laboratory of the University of Bremen for lending us the magnetic globe and Emma Goult for processing our survey results. We thank Rosella Nave, an anonymous reviewer, Faye Nelson and editor Solmaz Mohadjer for their comments that have improved the paper.

## 7 Data availability

All quiz answers are provided in Supplementary File 2.

## 8 Author contributions

All authors participated in the design and execution of the project, and all authors were involved in the writing of the manuscript.

## 9 Competing interests

The authors declare that they have no competing interests.

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
