# Peer review of "Magnetic to the Core – communicating paleomagnetism with hands-on activities"

_Geoscience Communication, 2021_

## Author Response (AR1)

**Reviewer 1:**

This manuscript describes an exhibition called 'Magnetic to the Core' in which the authors designed a public communication space to allow interactive learning to take place through a range of activities allowing people to learn about paleomagnetism. Overall I enjoyed this paper and felt that the authors covered useful information for others wishing to develop a similar approach to communicating a potentially unfamiliar topic to publics. I commend the authors for highlighting the extra time and financing necessary that can be taken for granted. The level of preparation and development for the outreach activities as well as the training session was fantastic to see, as well as the extensive piloting undertaken prior to the 'live' exhibition. Allowing input from others external to the project and providing a suitable approach and best practices in potentially complex scenarios is desirable so good to see this undertaken.

It is clear that the paper has considered both how to educate and engage the general public on an unfamiliar topic, something that can require a lot of effort if highly technical. The authors clearly described the activities, how they connected to each other and what each represented as part of the learning outcomes and key messages. The included activities were novel and well-designed allowing publics to easily grasp scientific concepts with these clearly presented in the paper.

In order to measure the learning outcomes and key messages that the authors were trying to share with their audience they asked people to complete a quiz to assess their level of knowledge (including those who had not visited the stand). The results from this showed a correlation with increased time and knowledge of visitors. They do acknowledge that groups also answered, however, so it is hard to distinguish if these groups (as opposed to an individual respondent) may have resulted in increased knowledge regardless of time spent at the stand. Nevertheless it is a clear and well-written paper with clear aims and should provide others with an example of successful science communication.

*We thank the reviewer for their kind words.*

Some minor points specific to the manuscript content:

 Abstract:

Line 24 – the authors state a 19.1% increase score but it is not clear what they mean. Change to 'increase in knowledge score' to clarify.

*Changed*

Introduction:

Line 33 – Although the authors go onto describe what paleomagnetism is later on in this section (or you can figure it out based on topics mentioned) it would be useful to provide a sentence at the start to briefly describe specifically what this is for non geoscientists.

*We have added "Paleomagnetism is the field of research that deals with reconstructing Earth's magnetic field in the past, as recorded by rocks" as the first line of the introduction.*

Outreach team & Training:

Line 110 – typo 'theme engaged'

*Changed*

Although the authors do include a paragraph (line 253) outlining ethics this is not an explicitly marked section as requested by the journal.

*We marked a section 2.5.1 Ethics*

Impact on knowledge of visitors:

Line 330 – the authors collected qualitative data and assessed to what extent the learning outcomes had been measured and clearly show this in Fig 9. If possible it would be interesting to see if there were any other key themes that came out in this data as over half of the responses did not relate to the outcomes (e.g. thematic analysis see Braun and Clarke, 2008).

*We agree with the reviewer that a more thorough thematic analysis (Braun & Clarke 2006?) of 'one thing you learned' would be possible, but as this is outside of our field of expertise and we have not performed a thorough thematic analysis before, we are a bit hesitant to do this. We are also not confident it would add a lot of insight to our outcomes. We have added a reference to all the answers in supplementary file 2, "A common theme emerging from the answers not linked to a learning outcome is the deep Earth (e.g. "The Earth has a solid core", "Liquid outer core")." And "Thematic analysis (Braun & Clarke, 2006) can be used to evaluate answers from this type of open question" to section 4.1.*

**Reviewer 2 (Rosella Nave)**

The paper present a very interesting science communication approach, particularly suitable for Science Festival. The methodology described and starting with the relevant trainig as conceived and carried out by the team, coulld be a reference for other Science communication activity, regardless of the topic and its public familiarity.

Perhaps a new stand design booth with fewer exhibits might be easier to manage both by the team and the public. I consider the proposed communicative approach and the whole methodology described, particularly relevant to be proposed to schools and teachers. The total cost of the exhibition was quite high, and this aspect could limit its organization by schools or in smaller science festival.

*We thank the reviewer for their kind words.*

Only one remarks to the text:

Abstract:

The relevant remark on decline number in Earth science students, that is suggested as one reason to developed an exhibits able to involve public in a scientific unfamiliar topic, should be the abstract incipit. The authors could give details on the seleted peculiar topic: palemagnetism.

*We have added "Paleomagnetism is the field of research that deals with reconstructing Earth's magnetic field in the past, as recorded by rocks" as the first line of the introduction.*

Introduction

line 61, 62 and 64: Three consequent senteces start with "The exhibition..."

*We have changed these sentences to "The exhibition is held in the building of the Royal Society in central London, close to Buckingham Palace. It attracts thousands of visitors each year, and runs for 7 days, at least 8 hours each day, and hosts 22 stands."*

**Comment from Faye Nelson**

Compliments to the authors and the Magnetic to the Core team. As stated in the article, the goal is to present Earth science/paleomagnetism outreach activities that can "easily be adapted and emulated to fit other countries, labs and audiences" (lines 75-76). We have had great success trying out the 'Rock or Choc' activity here in Aotearoa New Zealand and have used it with preschool and primary school groups, as well as with conference attendees at a family-friendly afternoon tea/outreach event. In the preschool setting some te reo Māori (Māori language) has been incorporated into the activity. The availability, portability and robustness of the Bartington MS2 magnetic susceptibility meter makes 'Rock or choc' a particularly simple/accessible outreach activity to carry out. We have not done any formal evaluations, but our observations support article's findings that 'Rock or choc' is popular will all ages. 'Rock or choc' has inspired us to develop additional outreach activities using some of the other MS2 probe attachments.

The 'Rock or choc' learning outcome that "Different materials have different magnetic properties" is a key piece of science knowledge/curriculum. It's interesting that only 7/210 of your respondents mentioned this as 'one thing...learned from the stand' (section 3.1/Figure 9), especially when 'Rock or choc' was the most popular activity. Is this because it was considered common knowledge in comparison to the other options? In our experience with 'Rock or choc' (where it was the sole activity), children as young as five years old were questioning why.  We are working on a simple graphic (sign for the table) that shows what's going on with diamagnetic vs ferromagnetic materials.

*We agree that the lack of reference to learning outcome 3 could have been because it is considered common knowledge by the participants. We have added "Although Rock or Choc was extremely popular, visitors do not often refer to learning outcome 3, possibly because it is considered common knowledge. We do note that there are many answers along the lines of "Chocolate Can Look Like A Rock", but we did not count these in our analysis, because they do not specifically mention magnetism."*

Thank you again for sharing the Magnetic to the Core activities with the paleomagnetism community.

(Just a small note on a typo: Figure 3 caption – Chock should be Choc.)

*Changed.*

**Comments from the editor**:
Great work, thanks for the submission. With some edits, this manuscript would be a great fit for publication in GC. My edits are shown below. I look forward to seeing the revised version:

Line 14 (abstract) - Can you provide some numbers to support this sentence: "Throughout recent years, there has been a decline in the number of Earth science students in the UK" For example, define "recent years", "decline" and "Earth science students" (undergrads, grads, etc.).

*This is elaborated in lines 45-50. We have added "The number of A-level students roughly halved since the 1980's (peak at nearly 4000 entries), with a recent drop from 2500 to 1500 from 2014 to 2018." And "From 2017 to 2019, the number of geology students at UK universities decreased around 10% yearly"*

Line 21 - Hands-on experiments or activities? There's a difference.

*Changed to hands-on activities everywhere.*

Introduction (section 1) - I suggest including a few more references to previously published education/outreach work related to paleomagnetism (in and outside the UK). For example, look on https://serc.carleton.edu/ for magnetism lesson plans that have gone under a peer-review process.

*Unfortunately searching for 'paleomagnetism' in this website does not bring up any peer-reviewed outreach work (only a paleomagnetic lab*
https://serc.carleton.edu/NAGTWorkshops/petrology/instruments/3094.html
*). We also searched on Scopus for paleomagnetism and outreach but this also turned up nothing. We know that outreach in paleomagnetism is done, but there just does not seem to be anything published, as mentioned in lines 56-58. Ayala et al. (2021) also note "There are creative ways to engage with people on planetary magnetism and its relevance to daily life, but an organized effort is lacking." We have added a reference to the study of Ayala et al. (2021).*

Line 38 – Please provide a reference for this sentence: "There is, however, little awareness of paleomagnetism and its uses and benefits amongst the UK public." Has this been documented?

*No, this is judged from the responses of visitors to our outreach activities and the fact that paleomagnetism is lacking from the curriculum.*

Line 75 - How can these activities be adapted for implementation in low-income countries or institutions with limited access to the kind of budget and equipment this outreach work requires? Please give an example.

*We have added this in section 2.3.3 "Alternatively, this activity can be carried out using a compass, which will show deflection of the needle when brought close to a magnetic rock pebble. The chocolate pebbles will not show a deflection of the compass needle."*

Line 85 - What can you do to increase the racial diversity of your team? I suggest adding some suggestion and perhaps include a reference to help other researchers with addressing this issue.

*There is a well-documented racial diversity crisis in geosciences in the UK (e.g. Dowey et al., 2021) and paleomagnetism internationally (Ayala et al., 2021). We have modified the text of section 2.1 "We formed a team that was diverse in terms of age, career stage and gender. The team consisted of PhD students (7, of which 2 women), postdocs (7, 4 women), with the addition of MSc students (2,*

*both women), as well as a technician (a man), a research fellow (a man), senior lecturers (2, both women) and professor (a man). Actions such as using name badges with the option to include preferred pronouns would be one way at future events to publicly recognise gender as non-binary and therefore increase the inclusivity of the event with respect to gender identity. Unfortunately, the lack of racial diversity in Earth sciences (Bernard and Cooperdock, 2018; Dowey et al., 2021; Dutt, 2019), and paleomagnetism in particular (Ayala et al., 2021) was also reflected in our team. In order to try and recruit team members from non-white backgrounds, we contacted other paleomagnetic laboratories in the UK several months before the exhibition, and tried to enthuse (PhD) students and researchers to join our team. Unfortunately this yielded no results in terms of increased racial diversity, and we were thus unable to form a team that showed the same diversity as the UK society and the audience at the exhibition. This is problematic as when young people do not see paleomagnetists, or scientists in general, who look and sound like them, then this gives the indirect impression that paleomagnetism is not for them. Evidenced action lists to address the lack of diversity in geosciences, including marginalization related to gender identity, ethnicity, sexuality, physical ability, socio-economic background and intersectionality, should be consulted when putting together teams that deliver outreach activities and the way the team and the resources they create are presented (e.g. Ali et al., 2021; Dowey et al., 2021; Vander Kaaden et al., 2021; Núñez et al., 2020)."*

New Table - Please add a table of activities showing at least 3 columns (name of the activity, learning model/pedagogical approach used, and the related learning objectives/outcomes). These are explained in the manuscript text but will read better as a table.

*We are not entirely sure what is meant here. We have added the following text: "Most of the hands-on activities benefit visual and kinesthetic learning preferences (https://vark-learn.com/). The interaction with team members at the stand catered for visitors with an aural learning preference (discussions and stories). The backdrop of our stand catered to visitors who had a learning preference for reading (see also Table 2)." As well as Table 2 at the end of section 2.3*

| Name of activity | Key message | Learning outcome | VARK |
|---|---|---|---|
| Licence to drill | 2 | 4 | V,K |
| Kickstool magnetometer | 2, 3 | 1, 2, 4 | K |
| Rock or choc | 2 | 3 | K |
| Magnetic globe | 1, 3 | 2 | V |

**Table 2: overview of relation between hands-on activities, key messages, learning outcomes and learning preferences (VARK: Visual, Aural, Read/Write, Kinesthetic)**

Line 185 - Can you provide a link (or reference) to Magnetic Measurements? This may help those interested in developing similar outreach activities by connecting them with relevant resources.

*http://www.magnetic-measurements.com/ added in line 193.*

Figure 4 - Do you have a video of Figure 4, and if so, can the link be shared in the text? This would help the readers better understand what you describe in section 2.3.4

*Unfortunately we do not have a video of this.*

Line 251 - Provide a link to the QuickTapSurvey app.

*https://www.quicktapsurvey.com/* added to line 269.

Fig 7 and 8 - Are these results statistically significant? What statistical approach did you use to assess this? This should be part of your data analysis methodology. Also add this info to your abstract.

*We have merged Figures 7 and 8 and added Table 4. We have modified the text as follows: "Because the data do not show a normal distribution, we use median values instead of averages. In total, there were 382 quiz responses recorded, of which 136 were by school-aged participants (see Table 4). As shown in Figure 7, the median quiz scores increased by 11.1% for those who visited the stand for less than 10 minutes compared with those who had not yet visited the stand. For those who visited the stand for more than 10 minutes, their median scores increased by 22.2%. Comparatively, visitors of school-age who visited the stand for an extended period of time experienced a more significant increase in the accuracy of their scores in the quiz (see Figure 7). School-age participants who visited the stand for less than 10 minutes achieved a 22.3% increase in median scores compared to school-age participants who had not yet visited the stand. School-age participants who visited the stand for more than 10 minutes experienced a 33.5% increase in their median scores compared to those who had not yet visited. We use the Wilcoxon rank sum test for common medians (also known as Mann-Whitney test). Comparing the responses from people who had not yet visited the stand to people who had visited <10 minutes yields Wilcoxon rank sum p-values of $p=0.010$ for all respondents, and $p=0.003$ for school-aged respondents. Comparing the <10 min responses to the >10 mins responses gives the following p-values for the tests: all respondents, $p<<0.001$, school-aged respondents, $p=0.004$. In both cases, at better than the 0.5% confidence level, we can reject the null hypothesis that the responses from the different groups come from distributions with the same median. Thus, the differences are significant. Performing a 2-sample t-test (for equal means) on average scores (which ignores the non-normality of the data) also shows there is a significant increase in quiz score." We have adjusted the numbers in the abstract and conclusions and added the word 'significant'.*

Line 332 - What about the remaining 121 answers? What were these about? I'm curious if they can be further categorized to reveal more insight into participants' learning.

*These are all provided in the supplementary information. We have added (see supplementary file 2) in line 334. and "A common theme emerging from the answers not linked to a learning outcome is the deep Earth (e.g. "The Earth has a solid core", "Liquid outer core")."*

Figure 9 - It would be helpful to give 1 example answer per learning outcome, either as part of this figure or in the manuscript text.

*We have added this to section 3.2*

Line 345 - Consider revising this sentence "Our results clearly show that our stand had an impact on visitors' knowledge regarding paleomagnetism and Earth's magnetic field". I don't think your data support this sentence. You results may indicate that, but they don't clearly show. For example, what you state in line 350 (i.e., participants may represent a well-educated portion of the population) is contradictory to this statement.

*We have changed 'clearly show' to 'indicate'.*

Section 4.2 - This section is a bit vague. How were these benefits documented? For example, how did the research group benefit from the exposure of your research to the public and the policymakers? If you can clearly document these benefits, this work has a real chance to inspire other researchers to engage in science outreach. I believe this is what this study is aiming for.

*This is anecdotal evidence, we asked our team members how they felt they benefited from the experience, and this statement reflects their answers. We did not do a formal survey. The comment from Faye Nelson suggests that our paper has already inspired other researchers to adapt outreach developed within the Magnetic to the Core project.*

Table 2 (budget) - Please include the hidden costs (e.g., volunteer work) and in line 397 insert the total budget amount for this project after "relatively small budget". Also, I suggest removing "relatively small" since for some institutions, this budget could be significant.

*Added (£24,000) to line 418.*
*Added "Although it is difficult to estimate the true costs that takes into account any hidden costs, just the number of people-hours at the stand during the exhibition is around 450. This does not include any preparations, travel, build-up and taking down of the stand. We estimate that taking into account all hours that all team members worked in total would probably put the true cost at double the cost that our budget shows."*

---

## Author Response (AR2)

Line 54 – Define "A-level" student or remove. Not everyone knows what this means.

*Added '(A-levels are UK secondary education qualifications, typically taken prior to university entrance)'*

Section 2 – Is it possible to do something similar to what you have done in lines 251-253 for other sub-sections in section 2? This would help educators from low-income countries with adapting your activities.

*We have added 'A low budget-version of this activity can be made using a bar magnet and a set of compasses positioned around the bar magnet, although it would not be possible to image the quadrupolar state this way.' Unfortunately we cannot suggest how to build a kickstool magnetometer because that would require an entirely new paper.*

Section 4.2 (Benefits to the Team) – Please delete this section (or rewrite and include evidence/examples). It's too general, does not add anything new, and is not backed up by evidence, qualitative or quantitative.

*Deleted*

Figure 8 – Are the added texts in Fig 8 examples of participants' responses? If so, mention that in the caption, and remind readers what the numbers are (1,2, …). I even suggest included the learning outcomes next to the numbers in the figure caption, so the readers don't have to refer to the manuscript text to understand your figure.

*No, these are our learning outcomes, as mentioned in the caption. They are also numbered in the figure. We have now repeated the information in the picture in the caption by adding 'Learning outcome 1 – Rocks are cool and they can record magnetic fields. Learning outcome 2 – The magnetic poles of the Earth reverse. Learning outcome 3 – Different materials have different magnetic properties. Learning outcome 4 – Paleomagnetism is practical, hands-on and anyone can do it.'*

Line 517-518, Revise this sentence: "We show that visiting the stand had a significant impact on visitors' knowledge of…" to something like "Our assessment data show a significant (above 95% ??) increase in …". I don't think you are measuring "impact" here, only changes in responses, correct?

*Changed to 'We show that visiting the stand had a measurable influence on visitors' knowledge of paleomagnetism and Earth's magnetic field through a significant increase in quiz scores.'*